# The mediating role of workplace milieu resources on the relationship between emotional intelligence and burnout among leaders in social care

Anna Kozák[1,2]*, Réka Schutzmann[1,3], Klára Soltész-Várhelyi[4], Fruzsina Albert[1]

**1** Mental Health Sciences Division, Semmelweis University, Budapest, Hungary, **2** Health Services Management Training Centre, Semmelweis University, Budapest, Hungary, **3** Archiescopical College of Veszprém, Veszprém, Hungary, **4** Institute of Psychology, Pázmány Péter Catholic University, Budapest, Hungary

* kozak.anna@emk.semmelweis.hu

## Abstract

**Data Availability Statement:** All relevant data are within the article and its Supporting Information files.

### Background

This study investigated the connection between emotional intelligence and burnout through the mediating role of workplace milieu resources (a sense of community and mutual trust between employees) among social care leaders in Hungary utilizing the Job Demand-Resources model as a reference. The study evaluated emotional intelligence across three dimensions: understanding our emotions, understanding others' emotions, and positive emotional appraisal.

### Methods

A cross-sectional and quantitative study was conducted from 11th April to 30th November 2019 targeting Hungarian social care leaders. Participants (N = 547) were recruited non-randomly trough a training organized for them. Data collection involved sociodemographic questions, the Assessing Emotions Scale and the Copenhagen Psychosocial Questionnaire. Descriptive statistics, Pearson's correlation, and three saturated serial mediations (ML with percentile bootstrap) were implemented. During the mediations, one dimension of emotional intelligence was used as a predictor in each model with the sense of community and mutual trust as serial mediators, and burnout as the outcome.

### Results

The results confirmed the role of the leader's emotional intelligence in creating a supportive workplace atmosphere and its indirect effect on burnout through these workplace milieu resources, while its direct effect was found not to be substantial. Among the three aspects of emotional intelligence, positive appraisal had the largest effect on burnout.

**Funding:** The author(s) received no specific funding for this work.

## Conclusions

This study suggests that emotional intelligence' influence on burnout is mediated by workplace milieu resources. Therefore, it is crucial to encourage leaders to use their emotional intelligence to create a positive emotional atmosphere rather than solely concentrating on emotional comprehension.

## Introduction

This research offers novel insights into the relationship between emotional intelligence (EI) and burnout, focusing on social care leaders in Hungary—a group particularly vulnerable to burnout due to unique organizational and bureaucratic pressures in Eastern Europe. Given the challenges associated with burnout, especially in helping professions [1, 2], and the effect of leaders' burnout on employees [3], this study highlights the potential of emotional intelligence and workplace milieu resources as mitigating factors. Utilizing the Job-Demand Resources model (JD-R) [4–10], the research explores how EI dimensions (understanding our own emotions, understanding others' emotions, and positive emotional appraisal), directly and indirectly influence burnout, focusing on the mediating role of workplace milieu resources-expressly, a sense of community and mutual trust. Key questions include whether EI influences burnout among social care leaders and whether workplace milieu resources can mediate this relationship. Moreover, this study also seeks to differentiate between the influence of different EI dimensions.

## Theoretical background

The literature review aims to synthesize research on workplace-related burnout, emotional intelligence, sense of community, and mutual trust between employees within the framework of the Job-Demand Resource model.

Since the mid-1970s, burnout research has spanned various occupations globally. The concept, however, has deeper roots, as issues related to the individual-work relationship have long been acknowledged. Freudenberger introduced the term "burnout" in 1974 [11], with Maslach concurrently applying it to helping professionals [12]. Despite over five decades of research, the definition and measurement of burnout continue to be debated [13, 14]. According to one of the most widely accepted definitions by Maslach and Leiter (2016), burnout is characterized as a psychological condition that develops as a prolonged reaction to persistent interpersonal stressors in the workplace [15]. Burnout can increase mental health issues like depression and anxiety and physical problems such as frequent sickness absences, sleep disturbances, and even heightened risk of severe conditions, including cardiovascular diseases [4].

According to the Job-Demand Resources (JD-R) model, developed in the early 2000s [6–10], understanding burnout requires examining various aspects and characteristics of work [7, 8]. The JD-R model [6–10] posits that each occupation can be characterized by two main categories: job demands and job resources [9]. Job resources encompass the physical, psychological, social, or organizational aspects of a job that (1) facilitate the achievement of work goals; (2) help mitigate job demands and their related physiological and psychological costs; or (3) foster personal growth and development. In contrast, job demands are the physical, psychological, social, or organizational elements that necessitate sustained physical and/or psychological effort—cognitive and emotional—resulting in various physiological and

psychological costs [9]. The JD-R model explains burnout through two interconnected processes. The first process involves demanding work conditions that result in persistent overexertion, ultimately leading to exhaustion. The second process emphasizes how inadequate job resources undermine employees' ability to meet work demands, fostering withdrawal behaviours [10]. Research suggests that job demands have a more robust predictive effect on burnout than job resources [4], making job demands a critical priority in burnout prevention strategies [7]. However, the interplay between job demands, resources, and burnout is complex and multifaceted, with evidence showing that the sole presence of various job resources could also buffer work-related burnout [4, 16].

Personal resources are also important when utilizing the JD-R model [17]. These attributes contribute to an individual's resilience and ability to shape their environment positively, aiding adaptation to workplace demands. Emotional intelligence (EI) is considered an important personal resource [4] and could have a favourable impact on other job resources [18]. Emotional intelligence is defined as "the ability to monitor one's and others' feelings and emotions, to differentiate between them, and to use this information to guide one's thoughts and actions" [19, p. 189], and has been assumed as a potential factor to reduce burnout risk [20]. Research highlights an inverse relationship between EI and burnout: for instance, Gallo de Moraes et al. (2015) [21] documented that higher EI levels correlated with lower emotional exhaustion and depersonalization among critical care professionals, while Platsidou (2010) identified a similar negative correlation between EI and burnout among special education teachers [22].

Individuals with high emotional intelligence are considered more adept at managing emotions, enabling them to cultivate positive relationships and social connections with coworkers and contribute to a cohesive workplace society [23–25]. Emotional intelligence is suggested to positively correlate with various social job resources, including a psychological sense of community and trust [26–28]. Based on Sarason's original definition, a *sense of community* is the recognition of shared similarities with others, an understanding of mutual interdependence, a readiness to uphold this interdependence by offering to others what one hopes to receive in return, and a sense of belonging to a broader, reliable, and stable system [29]. A sense of community is crucial for fostering relationships, promoting cooperation, and reducing absenteeism and turnover [30]. It is also suggested that a sense of community is associated with employee trust and respect. *Trust* gradually builds in groups where members cultivate familiarity over time—particularly when they rely on each other to attain shared goals or desired outcomes [31]. While definitions of trust vary [32], Mayer et al. (1995) describe it as the willingness of one party to assume vulnerability to the actions of another is rooted in the belief that the latter will undertake a specific action deemed essential by the trustor despite the other party's lack of oversight or control over those actions [33]. Within a strong community, members experience a deep sense of trust, belonging, safety, and mutual care, cultivating a collective identity [34]. Both the sense of community and trust have been independently associated with mitigating employee burnout [35, 36].

## Research objectives and hypothesis development

To meet organizational goals effectively, leaders must understand their and employees' mental processes [37]. Consequently, leaders with high emotional intelligence can effectively motivate their team members to work together towards common goals, facilitate dynamic interactions, build trust, and boost productivity, creating a supportive and dynamic work environment [38]. Their capacity to cultivate feedback-rich environments is essential for managing workplace stress and preventing burnout [39]. Additionally, the comprehension and effective

navigation of emotions, both one's own and those of others are considered beneficial and indispensable at all stages of social work [40].

Research indicates a negative correlation between emotional intelligence and burnout, suggesting that EI may be a significant tool for prevention and intervention [20–22]. While there has been extensive research on work-related burnout among frontline professionals and social workers [41–44], studies examining the role of EI within social care leadership remain sparse, and there is also a shortage of research focused explicitly on burnout among social care leaders [45]. This is particularly concerning, as burnout is a significant challenge in human services professions [2, 41–44]. Johnson and colleagues [1] found that social service work ranked among the six leading professions with the worst physical and psychological conditions and lowest job satisfaction compared to 26 other professions. Even though social care leaders usually do not work directly with clients, they are also highly susceptible to burnout [41].

Therefore, this research investigates the complex relationship between emotional intelligence and burnout, focusing on how EI directly and indirectly influences burnout. Emotional intelligence can be viewed as both a one-dimensional and a multifaceted concept. The study evaluated emotional intelligence through three distinctive dimensions, assuming that this approach would reveal a more nuanced and sophisticated relationship between EI and burnout. Building on the Job Demands-Resources model, the study proposes a conceptual framework in which the components of emotional intelligence are identified as predictors, burnout as the outcome, and workplace resources as mediators (as illustrated in Fig 1). In particular, it highlights the mediating role of workplace resources—such as a sense of community and mutual trust—in the connection between emotional intelligence and burnout among leaders in the social care sector.

Through exploring how EI, directly and indirectly, affects burnout and its influence on workplace milieu resources, this study examines whether emotionally intelligent leaders can effectively create environments that alleviate burnout. Additionally, it seeks to determine if specific EI components are particularly effective in enhancing these workplace resources, thereby offering a nuanced understanding of how EI contributes to workplace well-being. Addressing these questions will contribute to the literature by highlighting direct and indirect pathways through which emotional intelligence impacts burnout and identifying essential EI facets that foster a supportive workplace culture in social care settings. These findings could inform tailored social policy interventions to improve social care leaders' mental health and working conditions, contributing to the broader understanding of effective burnout prevention strategies.

## Hypotheses

The following hypotheses form the backbone of the present research model (see Fig 1) and will be tested separately using components derived from EI.

Multiple studies provide evidence supporting the positive effect of emotional intelligence in buffering burnout [20–22, 46]. Emotional intelligence allows individuals to identify and manage their emotions [19, 46–48], which could be especially important in emotionally demanding roles such as social care leadership. Therefore, the first hypothesis is as follows (**H1) Emotional intelligence negatively affects burnout** (H1 is tested with the correlation analyses).

Individuals with high emotional intelligence are more adept at creating and maintaining positive work environments and resourceful work cultures. Leaders hold a unique position in the workplace, as their decisions directly impact the social dynamics and environment. Emotionally intelligent leadership can benefit peoples' social relationships and could result in nurturing a positive workplace atmosphere with a sense of community, which could also promote

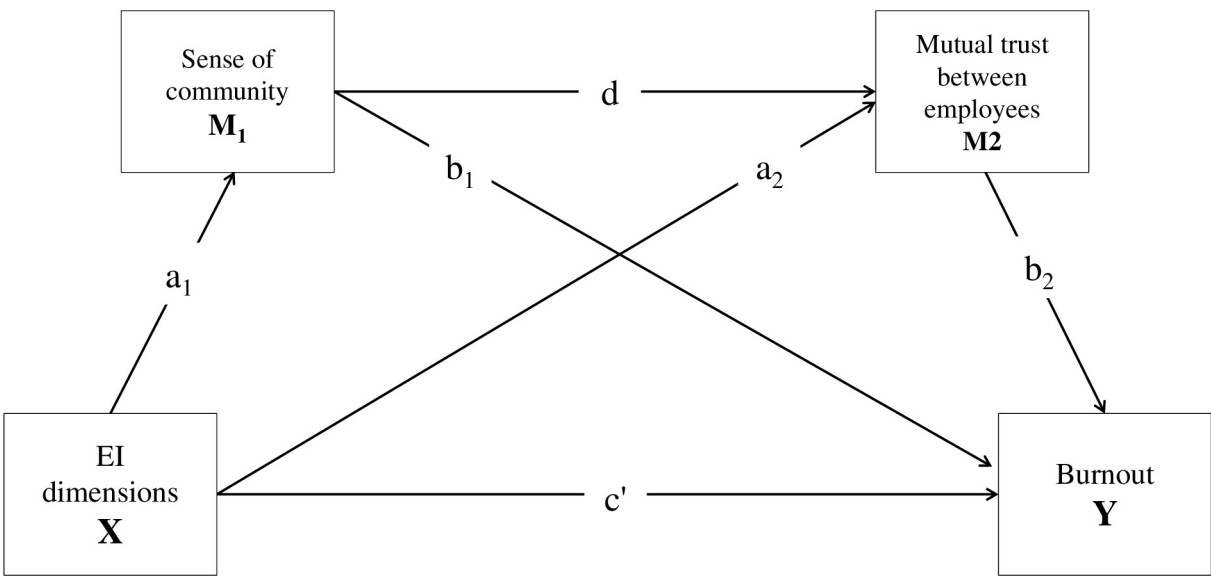

**Fig 1. The conceptual model of emotional intelligence and burnout with sense of community and mutual trust between employees as mediators.** Age and gender were used as control variables.

employee trust [23–28, 34–36, 49]. Accordingly, the model of this study assumes that a leader's emotional intelligence directly influences the cohesiveness of the community and indirectly fosters mutual trust among employees.

The second hypothesis is accordingly (**H2a) Emotional intelligence positively influences a sense of community** (path a1 on Fig 1) **and mutual trust between employees** (path a2 on Fig 1). (**H2b) The sense of community mediates the relationship between emotional intelligence and the mutual trust between employees** (indirect path a1*d on Fig 1).

A supportive community could provide emotional security and belonging, promote the emergence of trust between employees, thus increasing the likelihood of cooperation and mutual assistance. All of this could help employees cope with stress, and prevent burnout [26, 31, 35, 36, 50]. Therefore, the third hypothesis states the following (**H3) Workplace milieu resources negatively affect burnout** (path b1 and b2 on Fig 1).

The JD-R model proposes that various job resources could act as buffers against job demands and burnout [4–10, 16]. Individuals with high emotional intelligence can effectively foster social job resources such as a sense of community and trust, thereby contributing to a well-functioning, enriching work environment [26, 27]. Consequently, emotionally intelligent leaders may be more likely to leverage workplace resources, reducing burnout. Therefore, the fourth hypothesis posits that (**H4) Workplace milieu resources mediate the relationship between emotional intelligence and burnout** (indirect paths a1*b1, a2*b2 and a1*d*b2 on Fig 1).

## Materials and methods

### Research design and sampling procedures

A quantitative, cross-sectional primary study was conducted to unfold the correlations and mediational paths between emotional intelligence and burnout through workplace milieu resources. The target population were the middle and senior managers of the Hungarian social care system. The sampling was non-probability and purposive: participants in the survey were

recruited from the Master's Program of the Hungarian Social Service Management Training. Selection criteria required participants to be enrolled in the training and occupy leadership positions in the Hungarian social care system. The recruitment period spanned from April 11[th] to November 30[th], 2019 [51]. A comprehensive, multifaceted questionnaire was distributed to 667 social care leaders at Semmelweis University in Budapest, Hungary. The questionnaire was in Hungarian, and participants were asked to complete it on paper after their training sessions. The current study only analyses a part of the collected data.

## Ethics statement

The research was approved by the research review board of Semmelweis University in Budapest, Hungary (Ethical Approval of Research Semmelweis University Regional and Institutional Committee of Science and Research Ethics SE RKEB: 61/2019). During data collection, all adult participants were provided with a written consent form at the commencement of the questionnaire. Participation was voluntary and anonymous. Participants could withdraw at any time without consequences. They were informed that the processed results, containing anonymized individual data, would be published in scientific and educational publications.

## Participants

To determine the required sample size for a serial mediation with two mediators, Power Analysis [52] was performed with the following settings: $\alpha = .05$, target Power = .80, and weak or moderate correlations were assumed between the variables (r = .10 for X-Y, X-M2 and M1-Y; r = .20 for X-M1 and M2-Y; and r = .40 for M1-M2). A sample size of 350 participants was found to be sufficient to detect the indirect pathway. The questionnaire was distributed to 667 social care leaders, of whom 547 participants with an average age of 45.7 (SD = 7.1 Min = 23; Max = 60) completed the questionnaire. Women were highly overrepresented in the sample (female = 84.8%; male = 15.2%). The sample is highly educated, only 2 individuals (0.4%) have secondary education as their highest education, 428 individuals (79.0%) have tertiary education, and 112 individuals (20.7%) have postgraduate education, specialized training or doctoral degree. 145 individuals (26.6%) are working in villages, 242 individuals (44.4%) are working in smaller cities, 99 individuals (18.1%) are working in cities with county rights or county seats and 59 individuals (10.8%) in the capital. The average work (full-time job) experience was 21.8 years (SD = 8.9, Min = 2; Max = 42), and the average leadership experience was 10.2 years (SD = 6.9 Min = 1, Max = 33). 155 individuals (28.7%) are middle managers, and 386 individuals (71.3%) are senior managers. They have an average of 54.7 subordinates (SD = 97.5) For the mediation analyses, we excluded those who had missing value in any relevant variable (considered MCAR), so the final sample size for these analyses was N = 471.

## Instruments

**The Assessing Emotions Scale.** Emotional intelligence was assessed using the Hungarian-validated version [53] of the Assessing Emotions Scale (AES) [54]. Schutte et al. developed the scale [54] based on the original Salovey and Mayer model [19], and it is one of the most widely used and accepted questionnaires measuring all hypothesized aspects of EI [55]. The questionnaire comprises 33 items and initially demonstrated a one-factor structure [53]. Each item in the questionnaire employs a 5-point Likert scale, ranging from (1) strongly disagree to (5) strongly agree. Sample items include "I am aware of my emotions as I experience them" and "I expect good things to happen." The scale's one-factor structure has faced criticism, and subsequent research suggests that it performs better on a multidimensional scale. For instance, Petrides and Furnham [56] proposed using four factors; Austin [57] identified three factors;

Gignac et al. [58] employed six factors, while Keele and Bell [59] identified four factors [55]. Therefore, we opted to assess a component structure that best suits the sample of social care leaders, providing a more comprehensive understanding of the impact of emotional intelligence.

Using Principal Component Analysis (PCA), five components were identified, of which three were relevant to the research (Table in S2 File). The first component was identified as the ability to understand and experience one's emotions; to understand and appraise one's feelings and sensations (EI-Self). In the current study, reliability of this scale was good, with Cronbach's $\alpha = .75$. The second component was populated with items describing positive attitudes: positive feelings, expectations, and ideas about work settings and colleagues. Therefore, this dimension could be defined as the positive use of emotions (EI- Positivity), with excellent reliability, with $\alpha = .80$. Finally, the third component was characterized by items describing the understanding and appraisal of others' emotions (EI-Others), with good reliability, $\alpha = .79$. These three EI components overlap with the various components and multifactorial solutions suggested since the original questionnaire was published by Schutte et al. [54]. EI-Self could be similar to the component *Appraisal of Emotions in the Self* presented by Gignac et al. [58]; and to the *Appraisal of emotions* suggested by Petrides [56], Saklofske et al. [60], Austin et al. [57] and by Keele and Bell [59]. EI-Others could be similar to the *Appraisal of emotions in others* extracted by Gignac et al. [58] and to *social skills or managing others' emotions* suggested by Petrides and Furnham [56], Saklofske [60], and Keele and Bell [59]. EI-Positivity could be similar to *positive utilization* suggested by Chan [61] and to *optimism/mood regulation* proposed by Petrides and Furnham [56], Saklofske et al. [60], Austin et al. [57] and Keele and Bell [59].

**The Copenhagen Psychosocial Questionnaire (COPSOQ II).**   Burnout, mutual trust between employees, and the sense of community were assessed by adopting the relevant items from the Hungarian version of the Copenhagen Psychosocial Questionnaire (COPSOQ) II middle version [62]. Originally conceived to investigate psychological well-being and health promotion in workplace environments, the Copenhagen Psychosocial Questionnaire (COPSOQ) was developed by Kristensen and Borg at the Danish National Research Centre for the Working Environment from 1995 to 2007. The questionnaire encompasses various domains and inquiries, including social capital, and burnout. It has been translated into 25 languages and is widely employed as a risk assessment tool. COPSOQ comes in three versions: COPSOQ I, II, and III, each with short, medium, and long iterations [63]. The medium version of COPSOQ II was validated in Hungarian in 2012 [62]. COPSOQ II has the advantage of being integrated into the JD-R framework [64]. The Hungarian version contains 28 scales with 92 questions, many of which are assessed using five-point Likert-type scales (standardized to a 0–100 scale), exhibiting satisfactory internal consistency [62]. Burnout was assessed by asking: How often have you felt worn out?; How often have you been physically exhausted?; How often have you been emotionally exhausted?; How often have you felt tired? According to the agreement that exhaustion is the central component of burnout [13], the burnout-related questions of COPSOQ II seem suitable to capture the main tendencies of the phenomenon. The questions on the sense of community (labelled as social community at work in COPSOQ II) were: Is there a good atmosphere between you and your colleagues?; Is there good cooperation between the colleagues at work?; Do you feel part of a community at your place of work? The following questions measured mutual trust: Do the employees withhold information from each other?; Do the employees withhold information from the management?; Do the employees, in general, trust each other. In the surveyed sample, the Cronbach's alpha values for all pertinent scales were deemed acceptable, measuring at $\alpha = .90$ for burnout, $\alpha = .68$ for the sense of community, and $\alpha = .74$ for mutual trust between employees (Questions used in the questionnaire are presented in S1 File).

## Data analysis

As a preliminary step, a Principal Component Analysis (PCA) with Kaiser Criteria and Direct Oblimin Rotation was conducted to explore the structure of the AES-HU questionnaire (results of PCA are presented in the in S2 File). After reporting the demographic data (frequency and percentage for the nominal and ordinal data; mean and standard deviation for the scale variables) and analysing the descriptive statistics of the scales (mean, standard deviation, skewness and kurtosis), Pearson correlations were carried out to measure the relationship between the three identified EI components, burnout, and workplace milieu resources from the COPSOQ II. In addition, three serial mediation analyses were carried out to investigate the effect of the EI components on burnout through the sense of community and mutual trust between employees. In each model, one of the EI components was the predictor, and burnout was the outcome variable, with sense of community and mutual trust between employees as mediator variables. The direct and indirect effects of the emotional intelligence scales were calculated both on the workplace milieu resources and burnout. Age and gender were introduced to the models as control variables. As shown in the conceptual diagram (Fig 1), the serial mediation models included only manifest variables calculated from the questionnaire responses (commonly called Path Analysis or the structural part of CB-SEM) and they were examined in a saturated model, which is one of the accepted approaches in the field of the current research [65, 66]. In the saturated model, the point estimates of the regression coefficients and the indirect effects are the same, regardless of which regression calculation method or structural equation modelling is used (in this study, for reasons of convenience, Maximum Likelihood modelling was used). Standard Errors and Confidence intervals were obtained by percentile bootstrap for regression coefficients and indirect effects as well (Lavaan syntax of the saturated mediation models can be found in S4 File). Normality for all relevant variables (EI components, burnout, a sense of community and mutual trust between employees) were met, Skewness values were between S = -0.920 and 0.127 and the kurtosis values were between K = -0.280 and 2.019 [67]. Details can be found in the Table in S3 File. All analyses were performed using JASP (0.18.3) [68].

## Results

Social care leaders rated their burnout at an average of 48.68 (SD = 21.27) on a scale of 0–100. Their sense of community was relatively high, but their perceived mutual trust value fell slightly short (Table 1). As expected, there was moderate positive correlation between the two workplace milieu variables, and they both showed positive correlations with the EI components and negative correlations with burnout. Surprisingly, however, only weak negative or no correlations were found between the EI components and the job-related burnout. Demographic variables (age and gender) showed no correlation with any of the variables, except that woman had slightly higher EI-others value (r = .19 p < .001) and older people perceived slightly higher community (r = .11 p = .020). (Means and standard deviations of the EI components and the correlations between them are not interpreted, as they arose from PCA Oblimin rotation.) Results of the correlation matrix suggested that further analyses were needed to unravel the possible effects behind the contradictory results. Therefore, three serial mediation models were carried out to explore possible suppression effects.

### Mediation analysis of EI-Positivity

Table 2 shows the results of the analysis including EI-Positivity as predictor variable, sense of community and mutual trust between employees as serial mediators and burnout as outcome variable. Age and gender were used as control variables. It was found that EI-Positivity

**Table 1. Means, standard deviations and Pearson correlations between EI components, burnout, community and trust.**

| Variables | M | SD | α | 1 | | 2 | | 3 | | 4 | | 5 | | 6 |
|---|---|---|---|---|---|---|---|---|---|---|---|---|---|---|
| **1** EIS | 0.00 | 1.00 | .75 | — | | | | | | | | | | |
| **2** EIO | 0.00 | 1.00 | .79 | .27 | *** | — | | | | | | | | |
| **3** EIP | 0.00 | 1.00 | .80 | .29 | *** | .32 | *** | — | | | | | | |
| **4** Community | 82.48 | 12.45 | .68 | .11 | * | .27 | *** | .32 | *** | — | | | | |
| **5** Trust | 65.76 | 17.38 | .74 | .13 | ** | .16 | *** | .12 | ** | .46 | *** | — | | |
| **6** Burnout | 48.68 | 21.27 | .90 | -.01 | | -.03 | | -.12 | ** | -.17 | *** | -.16 | *** | — |

Note. EIS: EI-self; EIO: EI-others; EIP: EI-positivity; Community: Sense of community; Trust: Mutual trust between employees. M, SD, and α are used to represent the mean, standard deviation, and Cronbach's alpha respectively. Values are Pearson correlation coefficients. N = 471.

* indicates $p < .05$

** indicates $p < .01$

*** indicates $p < .001$.

positively predicted the sense of community ($a_1$), and that sense of community had a significant positive effect on trust (d). The direct effect of EI-Positivity on trust ($a_2$) was not significant, but its indirect effect through sense of community was confirmed ($a_1*d$). Only a marginally significant negative direct effect was found of community on burnout ($b_1$), but the effect of trust on burnout ($b_2$) was significant and negative. The direct effect of EI-Positivity on burnout (c') was not significant, whereas when analysing the indirect effects, a significant negative serial indirect effect was found of EI-Positivity on burnout through the path of sense of community followed by mutual trust ($a_1*d*b_2$). In contrast, indirect pathways including only one of the two mediators ($a_1*b_1$ and $a_2*b_2$), were found to be non-significant. Based on these

**Table 2. Serial mediation model of EI-Positivity (EIP) and burnout with two mediators.**

| Path | Variables | β | B | SE | p | | CI95 | |
|---|---|---|---|---|---|---|---|---|
| $a_1$ | EIP—Community | .31 | 3.84 | 0.54 | **< .001** | *** | 2.67 | 5.00 |
| $a_2$ | EIP—Trust | -.03 | -0.51 | 0.75 | .497 | | -2.16 | 0.94 |
| $b_1$ | Community—Burnout | -.09 | -0.16 | 0.09 | .086 | | -0.35 | 0.04 |
| $b_2$ | Trust—Burnout | -.11 | -0.14 | 0.06 | **.028** | * | -0.27 | -0.01 |
| d | Community—Trust | .46 | 0.65 | 0.06 | **< .001** | *** | 0.53 | 0.78 |
| c' | EIP—Burnout **(Direct effect)** | -.08 | -1.67 | 1.01 | .099 | | -4.07 | 0.53 |
| | **Indirect effects** | β | B | SE | p | | CI95 | |
| $a_1*d$ | EIP—Community—Trust | .143 | 2.48 | 0.42 | **< .001** | *** | 1.75 | 3.28 |
| $a_1*b_1$ | EIP—Community—Burnout | -.028 | -0.60 | 0.36 | .095 | | -1.52 | 0.11 |
| $a_2*b_2$ | EIP—Trust—Burnout | .003 | 0.07 | 0.11 | .516 | | -0.10 | 0.40 |
| $a_1*d*b_2$ | EIP—Community—Trust—Burnout | -.016 | -0.34 | 0.16 | **.039** | * | -0.72 | -0.04 |
| | Total indirect effect | -.041 | -0.87 | 0.36 | **.015** | * | -1.80 | -0.21 |
| | **Total effect** | β | B | SE | p | | CI95 | |
| c | EIP—Burnout | -.12 | -2.54 | 0.97 | **.009** | ** | -4.74 | -0.50 |

Note. Age and gender were included as covariates. EIP: EI-positivity; Community: Sense of community; Trust: Mutual trust between employees. N = 471. β and B indicates standardized and unstandardized regression coefficients, respectively. The standard errors and the confidence intervals of the B-values were obtained by percentile bootstrap.

* indicates $p < .05$

** indicates $p < .01$

*** indicates $p < .001$. Bold formatting indicates significance.

**Table 3. Serial mediation model of EI-Others (EIO) and burnout with two mediators.**

| Path | Variables | β | B | SE | p | | CI₉₅ | |
|---|---|---|---|---|---|---|---|---|
| a₁ | EIO—Community | .26 | 3.20 | 0.56 | **< .001** | *** | 2.15 | 4.27 |
| a₂ | EIO—Trust | .03 | 0.59 | 0.75 | .426 | | -1.04 | 2.17 |
| b₁ | Community—Burnout | -.12 | -0.20 | 0.09 | **.023** | * | -0.40 | -0.02 |
| b₂ | Trust -Burnout | -.11 | -0.13 | 0.06 | **.031** | * | -0.27 | -0.01 |
| d | Community -Trust | .45 | 0.62 | 0.06 | **< .001** | *** | 0.51 | 0.75 |
| c' | EIO—Burnout (**Direct effect**) | .01 | 0.18 | 1.01 | .861 | | -1.90 | 2.35 |
| | **Indirect effects** | **β** | **B** | **SE** | **P** | | **CI₉₅** | |
| a₁**d | EIO—Community—Trust | .12 | 1.99 | 0.40 | **< .001** | *** | 1.32 | 2.86 |
| a₁*b₁ | EIO—Community—Burnout | -.031 | -0.65 | 0.31 | **.034** | * | -1.48 | -0.04 |
| a₂*b₂ | EIO—Trust—Burnout | -.004 | -0.08 | 0.11 | .456 | | -0.41 | 0.12 |
| a₁*d*b₂ | EIO—Community—Trust—Burnout | -.013 | -0.27 | 0.13 | **.048** | * | -0.56 | -0.02 |
| | Total indirect effect | -.047 | -1.00 | 0.32 | **.002** | ** | -1.83 | -0.37 |
| | **Total effect** | **β** | **B** | **SE** | **P** | | **CI₉₅** | |
| c | EIO—Burnout | -.04 | -0.82 | 0.99 | .408 | | -2.88 | 1.25 |

**Note**. Age and gender were included as covariates. EIO: EI-others; Community: Sense of community; Trust: Mutual trust between employees. $N$ = 471. β and $B$ indicates standardized and unstandardized regression coefficients, respectively. The standard errors and the confidence intervals of the $B$-values were obtained by percentile bootstrap.

* indicates $p < .05$

** indicates $p < .01$

*** indicates $p < .001$. Bold formatting indicates significance.

results, the significant negative total effect of the EI-Positivity on burnout is realized entirely through the indirect route including both community and trust.

## Mediation analysis of EI-Others

Table 3 shows the results of the serial mediation model including EI-Others as predictor. A significant positive effect of EI-Others was found on community (a₁); and community had a significant positive effect on trust (d). The direct effect of EI-Others on trust (a₂) was not significant, while its indirect effect through sense of community (a₁*d) was significant and positive. A significant negative direct effect of community on burnout (b₁) was found, and the effect of trust on burnout (b₂) was also significant and negative. The direct effect of EI-Others on burnout (c') was not significant. Furthermore, the indirect effect between EI-Others and burnout through the pathway involving community but excluding trust (a₁*b₁) was found to be significant. On the other hand, without community, no significant path through trust was found (a₂*b₂). In addition, the negative serial mediation through both mediators (community and trust) was also significant (a₁*d*b₂).

## Mediation analysis of EI-Self

EI-Self positively predicted the development of the sense of community (a₁), and a positive significant relationship was found between the two mediators (d). Both the direct and indirect effect of EI-Self on mutual trust between employees (a₂ and a₁*d) was found to be significant. Additionally, both mediators had a significant negative effect on burnout (b₁ and b₂). However, the direct effect of EI-Self on burnout (c') was not significant, suggesting that the relationship between EI-Self and burnout could only be established through the indirect pathways of the mediators. The combined effect of the indirect pathways between EI-Self and burnout was

**Table 4. Serial mediation model of EI-Self (EIS) and burnout with two mediators.**

| Path | Variables | β | B | SE | p | | CI₉₅ | |
|---|---|---|---|---|---|---|---|---|
| a₁ | EIS—Community | .11 | 1.32 | 0.57 | **.020** | * | 0.22 | 2.46 |
| a₂ | EIS—Trust | .08 | 1.40 | 0.71 | **.050** | * | 0.14 | 2.71 |
| b₁ | Community—Burnout | -.12 | -0.20 | 0.09 | **.021** | * | -0.39 | -0.01 |
| b₂ | Trust—Burnout | -.11 | -0.14 | 0.06 | **.030** | * | -0.27 | 0.00 |
| d | Community—Trust | .45 | 0.62 | 0.06 | **< .001** | *** | 0.51 | 0.74 |
| c' | EIS—Burnout (**Direct effect**) | .02 | 0.39 | 0.97 | .686 | | -1.61 | 2.22 |
| | **Indirect effects** | **β** | **B** | **SE** | **p** | | **CI₉₅** | |
| a₁*d | EIS—Community—Trust | .047 | 0.82 | 0.36 | **.022** | * | 0.05 | 1.57 |
| a₁*b₁ | EIS—Community—Burnout | -.013 | -0.27 | 0.16 | .100 | | -0.73 | 0.02 |
| a₂*b₂ | EIS—Trust—Burnout | -.009 | -0.19 | 0.13 | .145 | | -0.58 | 0.00 |
| a₁*d*b₂ | EIS—Community—Trust—Burnout | -.005 | -0.11 | 0.07 | .115 | | -0.36 | 0.01 |
| | Total indirect effect | -.027 | -0.57 | 0.23 | **.014** | * | -1.19 | -0.21 |
| | **Total effect** | **β** | **B** | **SE** | **P** | | **CI₉₅** | |
| c | EIS—Burnout | -.01 | -0.18 | 0.98 | .857 | | -2.21 | 1.60 |

**Note**. Age and gender were included as covariates. EIS: EI-self; Community: Sense of community; Trust: Mutual trust between employees. N = 471. β and *B* indicates standardized and unstandardized regression coefficients, respectively. The standard errors and the confidence intervals of the *B*-values were obtained by percentile bootstrap.

* indicates *p* < .05

** indicates *p* < .01

*** indicates *p* < .001. Bold formatting indicates significance.

found to be significant and negative; however, none of the separate pathways of the serial mediations appeared strong enough to be significant (Table 4).

## Discussion

It is suggested that emotional intelligence could enhance individuals' ability to engage and cooperate openly with others, fostering emotional safety, a positive workplace atmosphere, and a sense of community. Such an environment can foster building trust among coworkers, which can help mitigate symptoms of burnout [26–28, 31–33, 35, 36]. Grounded in the JD-R model [4–10], this study hypothesised that emotional intelligence could protect against work-related burnout directly and indirectly among social-care leaders. Emotional intelligence was examined through three distinct dimensions, expecting that each would exert varying degrees of influence on burnout. The dimensions were as follows: EI-Self, EI-Others and EI-Positivity. EI-Self pertains to the understanding and recognising one's own emotions [56–60], and EI-Others refers to the ability to comprehend the emotions of others [56, 58–60], whereas EI-Positivity comprehends positive appraisal and utilisation of emotions [57, 59–61].

This research also proposed that the indirect effects of emotional intelligence dimensions on burnout, mediated by workplace milieu resources, would outweigh their direct effects. Specifically, two workplace milieu resources—a sense of community and mutual trust—were identified as potential mediators, highlighting leaders' pivotal role in fostering these social resources [36]. Consequently, four hypotheses were formulated to support the research model: **H1: Emotional intelligence negatively affects burnout. H2a: Emotional intelligence positively influences a sense of community and employee mutual trust. H2b: The sense of community mediates the relationship between emotional intelligence and mutual trust between employees. H3: Workplace milieu resources negatively affect burnout. H4:**

**Workplace milieu resources mediate the relationship between emotional intelligence and burnout.**

Contrary to the initial expectations of a straightforward link between emotional intelligence and burnout, this study revealed unexpected findings that challenge existing literature. While previous research suggested a significant role for emotional intelligence in mitigating burnout across different organizational contexts [20–22, 46], the results of this study indicate weak or negligible correlations between EI dimensions and burnout. Therefore, the first hypothesis **H1** was not supported. A meta-analysis on this topic [69] found the relationship between emotional intelligence and burnout to be generally moderate, with high heterogeneity among studies. In the current study, the observed correlations fall within the weaker range of this heterogeneity. The meta-analysis also suggested that the strength of the effect of emotional intelligence on burnout may be influenced by other factors, including the characteristics of the workplace environment. These findings indicate that emotional intelligence may only exert its protective effect in the right community, while, for example, in a stressful or hostile workplace environment, sensitive perception of other people's emotions can even be counterproductive. These results may suggest the possibility of complex suppression effects and indirect effects. Thus, the investigation of the relationship between emotional intelligence and burnout continued in the context of workplace milieu resources.

In the mediation models, all EI dimensions had a significant direct effect on the sense of community (a1), with only EI-Self demonstrating a significant direct effect on trust (a2). However, all three EI dimensions showed a significant positive indirect effect on trust through the sense of community (a1*d). This finding supports the hypothesis **H2a**, indicating that emotional intelligence positively influences the sense of community and confirm **H2b** indirectly impacts employee trust, aligning with existing literature [23–28, 34–36, 49]. Leaders hold a unique position in influencing the workplace environment and the overall atmosphere of the workplace community. They can shape the work process in such a way that it promotes cooperation. A positive and cooperative work environment can naturally foster trust among employees without direct intervention from the leader. Among the three EI dimensions, EI-Positivity yielded the strongest direct effect on the sense of community and the most pronounced indirect effect on trust, closely followed by EI-Others. Conversely, the effect of EI-Self on these workplace milieu resources was the weakest. This outcome confirms that a leader's positive attitude, optimism, and ability to understand others generally contribute to the effective functioning of the work community. As expected from the literature [35, 36], both workplace milieu resources had a negative direct impact on burnout (b1 and b2), supporting the third hypothesis **H3** of this study.

The direct effect of emotional intelligence on burnout did not show significance for any EI components, with only EI-Positivity having a marginally significant direct effect. On the other hand, the indirect effects passing through the workplace milieu resources were significant for all the EI subscales, thus supporting the fourth hypothesis **H4** of this study.

The total indirect effect of an EI dimension can be dissected into three paths: one path going through both workplace milieu resources (a1*d*b2), another path involving only the community and excluding trust (a1*b1), and a third path involving trust but excluding the community (a2*b2). Between EI-Positivity and burnout, only the indirect pathway including both the sense of community and mutual trust was significant. In contrast, indirect pathways including only a sense of community without mutual trust or only mutual trust without a sense of community were not significant. These findings suggest that the optimistic attitude of leaders does not provide sufficient protection against burnout in a workplace where only the atmosphere is good, but coworkers fundamentally do not trust each other or the management.

Similarly, a strong climate of trust without active cooperation is also inadequate in alleviating burnout symptoms, even when leaders exhibit a positive attitude.

The indirect effect between EI-Others and burnout containing both the sense of community and mutual trust was positive, similarly to EI-Positivity. Furthermore, the pathway going through the sense of community but excluding mutual trust was also significant, while the path involving only mutual trust but excluding sense of community was not significant. This discrepancy suggests that it may be more practical to use one's ability to understand others' emotions to cultivate a strong, community-oriented environment that indirectly fosters mutual trust. This approach could be less emotionally demanding than maintaining direct trust with each employee.

EI-Self seems to have the weakest explanatory power of the three components: for understanding our own emotions, none of the direct or indirect effects were significant when the various indirect paths were separated, and only the total indirect effect was significant. Increased self-awareness can mitigate burnout but may exacerbate burnout if the leader lacks the resources to handle work-related demands [70]. It might be possible that for EI-Self, a sense of community and mutual trust may not be the most effective resources for shielding against burnout. Hence, exploring additional resources to pinpoint the most suitable mediators is crucial.

In summary, the findings suggest that different aspects of emotional intelligence do not impact burnout through the same mechanisms and workplace resources. Regarding EI-Positivity, besides the significant indirect effect, also a marginally significant direct effect remained, indicating that a leader's positive, optimistic attitude in itself might reduce the likelihood of burnout. Additionally, a trusting work community, which is more easily formed with such a leader, also provides protection against burnout. On the other hand, for EI-Self and EI-Other, while there is a significant negative indirect effect on burnout, there is no negative direct effect.

Access to and understanding of emotions does not directly protect against burnout, but it does assist the leader in creating a supportive and trusting workplace community that can act as a buffer against burnout. It is noteworthy that the direct effect of these two EI scales is not only not significantly negative, but also moved into the positive value range, suggesting a potentially complex suppressive effect. While EI-Self and EI-Other help reduce burnout through social workplace resources, the continuous consideration of emotions can be tiring, highlighting the potentially dual nature of emotional intelligence, which explanation is in line with the interpretation of other authors [71], since being "on guard" and being "vigilant" of others emotion can burden the attention of the leader leading to exhaustion.

## Conclusions

The findings of this study support that emotional intelligence dimensions (EI-Self, EI-Others and EI-Positivity) can help cultivate workplace milieu resources that are inversely associated with burnout. The impact of the self-awareness dimension of Emotional Intelligence (EI-Self) on workplace milieu resources was found to be marginal. This nuanced finding indicates that while EI-Self may contribute to resource cultivation, its influence is less substantial than anticipated. In this unique context and sample, the limited impact of EI-Self emphasizes prioritizing the interpersonal aspects of emotional intelligence. Understanding and managing the emotions of others (EI-Others) and possessing a positive attitude (EI-Positivity) have a more profound impact on cultivating workplace milieu resources and preventing burnout. Social care leaders who excel in these areas can better navigate their teams' emotional landscapes, enhancing workplace milieu resources, collaborative practices, and trust. Future research and training

programs for social care leaders should prioritize these interpersonal aspects of emotional intelligence, ensuring that leaders can create and sustain emotionally healthy and resource-rich work environments. This comprehensive approach to emotional intelligence can lead to a more resilient and effective social care workforce.

## Theoretical implications and limitations

This study offers valuable insights into the relationship between emotional intelligence dimensions, workplace milieu resources, and burnout among Hungarian social care leaders, a group particularly vulnerable to burnout due to administrative and bureaucratic pressures. To our knowledge, this is the first research to explore these dynamics among social care leaders in Hungary, addressing a significant gap in the literature within Eastern Europe's unique organisational and political context. The findings extend the application of the Job-Demands Resources (JD-R) model by illustrating how emotional intelligence dimensions influence workplace milieu resources, amplifying emotional intelligence's protective effects in mitigating burnout. This research highlights how emotionally intelligent leaders can better navigate challenges, leverage workplace support, and foster trust and community to build leadership resilience in high-pressure environments by emphasising the mediating role of job social resources.

The study could also support strategic planning implications for decision-makers, advocating for targeted leadership development programs alongside emotional intelligence and burnout prevention training. These programs should equip leaders with the skills to create and sustain resource-rich environments that promote inclusivity, engagement, and emotional safety. Such initiatives could enhance leaders' and employees' well-being, improve job satisfaction, and strengthen organisational commitment, ultimately supporting a sustainable and resilient workplace culture that can lead to better organisational outcomes. Despite these contributions, several empirical gaps remain, emphasising the need for further research to validate and expand upon these findings.

**Firstly**, this study was based on a cross-sectional design. As a result, it cannot reflect the evolving dynamics and interactions between the studied phenomena, and reverse or reciprocal causal relationships cannot be analysed. To address these limitations, future research could employ a longitudinal design to examine changes and interactions among emotional intelligence, workplace resources, and burnout over time. However, the mediation model establishes a logical interpretation, enabling it to capture and examine the variables' relationships effectively. **Secondly,** the sample was unique due to gender, age, and Hungarian context characteristics, potentially impacting the correlation values between the variables. To overcome this limitation, all the data were controlled for age and gender. **Thirdly**, burnout and workplace milieu resources were measured using selected questions from the COPSOQ II questionnaire; however, more specific and targeted questionnaires should be considered to broaden the hypotheses and further extract the relationship between EI and burnout. Additionally, other job resources that can mediate the relationship between EI and burnout at the workplace should also be considered. Furthermore, to provide a comprehensive picture, demands should also be incorporated into the model, as, according to the JD-R model, resources are less strongly (negatively) associated with burnout than job demands [4]. **Fourthly,** the impact of cultural factors specific to Hungary and other Eastern European nations remains underexplored despite their potential significance in shaping leadership styles and organizational support structures. **Finally**, the recruitment period for this study (April 11th to November 30th, 2019) predates the global COVID-19 pandemic, which has drastically reshaped workplace dynamics, particularly in the social and healthcare sectors. Although Hungary largely resumed

in-person work—aligned with the personal nature of social work—the pandemic introduced profound challenges that could exacerbate burnout and workplace demands. For example, Dima, Schmitz, and Simon's (2021) research found elevated stress levels among Romanian social workers during the pandemic associated with increased burnout [72]. Whereas McCoyd and Colleagues suggested that concerning the COVID-19 pandemic, social workers demonstrated resilience, adapting to increased demands, integrating new technologies, and addressing service gaps despite experiencing symptoms of burnout [73]. These findings suggest that the structural and psychological impacts of the pandemic may have influenced the interplay between emotional intelligence, workplace resources, and burnout prevention among leaders in social care. A follow-up study incorporating post-pandemic data will address these limitations, focusing on workplace demands, structural changes, and the evolving roles of emotional intelligence and resources in mitigating burnout. This future research aims to understand better the social care sector's changing dynamics and practical strategies for leader support in a post-pandemic context.

Despite certain limitations, this analysis reveals significant implications for future research, highlighting the necessity of considering emotional intelligence in burnout management for social care leaders. Examining specific components of emotional intelligence to fully capture its nuances is crucial. While the JD-R model suggests that resources generally have a less intense predictive value for burnout, the mediation models of this study indicate that workplace milieu resources can significantly influence the relationship between emotional intelligence and burnout.

## Supporting information

**S1 File. Survey questionnaire items.**
(DOCX)

**S2 File. Results and loadings of Principal Component Analysis of EI items after Direct Oblimin rotation.**
(DOCX)

**S3 File. Skewness and kurtosis values of relevant variables.**
(DOCX)

**S4 File. Lavaan syntax.**
(DOCX)

**S1 Dataset.**
(XLSX)

## Acknowledgments

We sincerely thank the research team at Semmelweis University's Mental Health Sciences Division for their expert guidance and invaluable recommendations during this study. Additionally, we thank the anonymous Hungarian social service managers for their respectful and comprehensive completion of the questionnaire, drawing on their extensive knowledge.

We would like to declare that for this article, Google Translate, DeepL, and Grammarly, along with university proofreading services, were employed to translate short passages and enhance grammatical accuracy.

## Author Contributions

**Conceptualization:** Anna Kozák, Réka Schutzmann, Klára Soltész-Várhelyi, Fruzsina Albert.

**Data curation:** Anna Kozák, Réka Schutzmann, Klára Soltész-Várhelyi.

**Formal analysis:** Anna Kozák, Réka Schutzmann, Klára Soltész-Várhelyi.

**Methodology:** Anna Kozák, Réka Schutzmann, Klára Soltész-Várhelyi.

**Supervision:** Fruzsina Albert.

**Validation:** Anna Kozák, Réka Schutzmann, Klára Soltész-Várhelyi.

**Visualization:** Anna Kozák, Réka Schutzmann, Klára Soltész-Várhelyi.

**Writing – original draft:** Anna Kozák, Réka Schutzmann, Klára Soltész-Várhelyi.

**Writing – review & editing:** Anna Kozák, Réka Schutzmann, Klára Soltész-Várhelyi, Fruzsina Albert.

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
