## [Decision Letter · Decision Letter 0]

20 Sep 2024

PONE-D-24-29991The mediating role of workplace milieu resources on the relationship between emotional intelligence and burnout among leaders in social carePLOS ONE

Dear Dr. Kozák,

Thank you for submitting your manuscript to PLOS ONE. After careful consideration, we feel that it has merit but does not fully meet PLOS ONE’s publication criteria as it currently stands. Therefore, we invite you to submit a revised version of the manuscript that addresses the points raised during the review process.

We look forward to receiving your revised manuscript.

Kind regards,

Supaprawat Siripipatthanakul, Ph.D.,DBA, MS. (Management), DDS. etc.

Academic Editor

PLOS ONE

Journal Requirements:

3. We notice that your supplementary tables are included in the manuscript file. Please remove them and upload them with the file type 'Supporting Information'. Please ensure that each Supporting Information file has a legend listed in the manuscript after the references list.

Additional Editor Comments:

Please create four tables for each of the two columns. The left column is for reviewers' comments; the other is for the author(s)' revision (one-by-one comment and revision). Kindly carefully read and follow the comments. In case of disagreement (but please avoid this), you may explain why you argue and do not intend to follow the comments. Please highlight the revised statement in yellow. The academic editor's decision is revisions are required. Before resubmission, please check for plagiarism and similarity with AI-generated documents below 15%. Kindly check citations and references to see if they are following the PLON's regulations.

Reviewers' comments:

Reviewer's Responses to Questions

**Comments to the Author**

1. Is the manuscript technically sound, and do the data support the conclusions?

Reviewer #1: No

Reviewer #2: No

Reviewer #3: Partly

Reviewer #4: Yes

2. Has the statistical analysis been performed appropriately and rigorously? 

Reviewer #1: No

Reviewer #2: No

Reviewer #3: No

Reviewer #4: Yes

3. Have the authors made all data underlying the findings in their manuscript fully available?

Reviewer #1: No

Reviewer #2: No

Reviewer #3: Yes

Reviewer #4: Yes

4. Is the manuscript presented in an intelligible fashion and written in standard English?

Reviewer #1: No

Reviewer #2: No

Reviewer #3: Yes

Reviewer #4: Yes

5. Review Comments to the Author

Reviewer #1: The topic and keywords are acceptable. However, the abstract needs methods and analysis techniques, such as administering only surveys. The analysis uses SPSS and SEM, such as AMOS, Smart PLS, and ADANCO. The introduction could be more comprehensive. A literature review is necessary for defining terms including all variables in this study based on theories, and research hypotheses. The research objective (aim) is suggested to move to the introduction and add research questions. Research design is necessary for population, sampling technique, data collection, and data analysis (SPSS for descriptive analysis in frequency, percentage, and mean). The inferential statistical analysis of PLS-SEM is necessary to explain. For data analysis in Table 2, please add *significant level at p<0.05, **significant level at p<0.01, and ***significant level at p<0.001. Discussions are needed to follow the research hypotheses, and more tables are required for assumptions. Avoid using we, but use the researchers instead. When research hypotheses are unclear, discussions and conclusions are also not clear. The conclusion suggests how a strategic planner could improve the outcome based on predictors. 

Reviewer #2: Plagiarism and AI Writing similarities are significant for now and must be checked to avoid ethical issues. When using Grammarly to check AI generated document similarity was found that 49% of this manuscript is similar to AI-generated documents. Thus, the reviewer suggest the authors revise the manuscript to be plagiarism and AI-similarity to be 15% or less.

Reviewer #3: this research topic is interesting but unfortunately there is no explanation of the novelty that is highlighted so that it seems ordinary, especially since the introduction is only brief and has not been able to explain what problems will be solved and urgent to do research.

first suggestion, in the abstract there is no explanation of the methodology used, this is very important to be explained.

second suggestion, to add an in-depth discussion of the problems and empirical gaps that occur so that this research has good useful value

The third suggestion, make hypothesis development on each influence relationship with relevant articles that support the research framework, because this research uses quantitative methods so it is mandatory to make hypotheses and will be answered in the discussion section.

The fourth suggestion, in the methodology section, it is necessary to determine the sampling technique used, besides that to provide better value ... researchers need to add questionnaire statements in tabular form so that they can be easily read.

The fifth suggestion, in the research results section, there should be validity and reliability testing so that the data used can be relied on, at least the indicator part of each variable.

The sixth suggestion, in the discussion section, it is highly recommended to discuss each influence relationship so that the results and discussion can be interpreted properly.

The seventh suggestion, the conclusion should be first and then the implication, the conclusion is a summary of the discussion while the implication is the impact after this research is completed.

Translated with DeepL.com (free version)

Reviewer #4: Technically sound, the manuscript makes use of suitable statistical techniques such as principal component analysis (PCA), power analysis, and serial mediation analysis. The information backs up the conclusions, especially when it comes to showing how workplace milieu resources mediate the link between burnout and emotional intelligence (EI). The facts rationally support the nuanced conclusion that EI-Self has a lesser influence, which has been well-discussed. The Job Demand-Resources model and the study's emphasis on social care leaders contribute to the field's understanding, and the results successfully support the conclusions made.

The statistical analysis has been conducted rigorously. The power analysis ensured sufficient sample size for detecting effects, and PCA was correctly used to validate the structure of the EI scale. Pearson correlations and serial mediation models are suitable techniques for this type of study, and bootstrapping was appropriately applied for testing indirect effects. The authors also verified the normality of the data, ensuring the reliability of the mediation models. The inclusion of covariates like age and gender further strengthens the robustness of the findings.

Strengths: The study adds significant value by focusing on social care leaders, a population not widely studied in burnout literature. The detailed breakdown of emotional intelligence into components (EI-Self, EI-Others, EI-Positivity) provides a nuanced view of how EI impacts burnout through workplace resources.

Limitations: The cross-sectional design limits the ability to infer causal relationships. This could be addressed in future research by incorporating longitudinal designs to observe changes over time.

Ethics: The study followed appropriate ethical guidelines, with voluntary participation, informed consent, and anonymization of data.

No concerns about dual publication or research ethics were identified.

6. PLOS authors have the option to publish the peer review history of their article (what does this mean?). If published, this will include your full peer review and any attached files.

Reviewer #1: No

Reviewer #2: No

Reviewer #3: No

Reviewer #4: No

---

## [Author Response · Author response to Decision Letter 0]

22 Oct 2024

Dear Editor and Reviewers,

We sincerely appreciate your valuable comments and insights on our manuscript. We have thoroughly reviewed all your suggestions and made comprehensive amendments to enhance the document. We believe these revisions have significantly improved its quality. The changes proposed by the reviewers are detailed in the table format required in the PLOS ONE Revision letter.

Below are our responses to the journal's requirements:

1. **Manuscript Style Requirements**: We appreciate your feedback. We have thoroughly reviewed the style requirements once more to ensure our manuscript meets all guidelines.

2. **Data Sharing Policy**: We have established the database and will upload it to one of the recommended repositories upon acceptance of the manuscript.

3. **Supplementary Tables**: We have created new supplementary documents, and all tables and supporting information have been relocated there. Additionally, we have ensured that a legend list for all supplementary materials is included in the manuscript.

Authors Comments to the Reviewers

Reviewer1

Reviewers' Comment Authors revision

The abstract needs methods and analysis techniques, such as administering only surveys. 

-Thank you for your statement. We have extended and adjusted the abstract as required.

The analysis uses SPSS and SEM, such as AMOS, Smart PLS, and ADANCO. 

-Thank you for your statement. However, we would like to clarify that our article does not reference the tools or methods you mentioned. As the Data Analysis section outlines, the analyses were conducted using JASP. The data analysis section was adjusted to clarify that we used a saturated model of serial mediation. The parameterization of these analyses was also added to this section.

The introduction could be more comprehensive. 

-Thank you very much for your suggestion. We have extended the introduction and made it more comprehensive.

A literature review is necessary for defining terms including all variables in this study based on theories, and research hypotheses. 

-Thank you very much for your comments. We have entered hypotheses based on the assumptions and facts of the literature review. For emotional intelligence, we extended the definition with the main components of the construct as follows: "Nevertheless the difference between the theories the facets are similar because of the dominance of Salovey and Mayer's theory. The four more widely used facets or constructs are the following: "(1) perceiving emotions (in self and others), (2) regulating emotions in self, (3) regulating emotions in others, and (4) strategically utilizing emotions" [25, pp.3].."

For all concepts: EI, Burnout, JD-R mutual trust and sense of community, the definitions have been included in the text so far as follows:

Burnout: "One of the most widely accepted definitions of burnout is Maslach and Leiter (2016), which defines burnout as "a psychological syndrome that develops as a prolonged response to chronic interpersonal stressors at work" [13, p. 104] (Manuscript p. 5).

EI: "Salovey and Mayer defined emotional intelligence as "the ability to observe one's own and others' feelings and emotions, to distinguish between them, and to use this information to guide one's thoughts and actions" [24, p. 189] (Manuscript p. 5).

JD-R: Workplace resources include "physical, psychological, social, or organizational aspects of work that are either: (1) functional in achieving work goals; (2) reduce job demands and associated physiological and psychological costs; (3) promote personal growth and development" [30, 344. On the other hand, demands are "physical, psychological, social or organisational aspects of work that require sustained physical and/or psychological (cognitive and emotional) effort and therefore have certain physiological and/or psychological costs" [30, p. 344].

Sense of community: "According to Sarason's original definition, a sense of community is "a sense of similarity to others, a recognized interdependence with others, a willingness to maintain this interdependence by giving to or doing for others what is expected of us, and a sense of being part of a larger, reliable and stable structure"' [35, p. 157] (Manuscript p. 8.)

Trust: "Although the definition of trust is still debated [39], many researchers accept the definition proposed by Mayer et al. which defines trust as "the willingness of a party to be vulnerable to the actions of another party based on the expectation that the other party will perform a certain action important to the trust-giver, regardless of his or her ability to control or direct the other party" [44, p. 712] (Manuscript p. 8). 

Trust: "Although the definition of trust is still debated [39], many researchers accept the definition proposed by Mayer et al. which defines trust as "the willingness of a party to be vulnerable to the actions of another party based on the expectation that the other party will perform a certain action important to the trust-giver, regardless of his or her ability to control or direct the other party" [44, p. 712] (Manuscript p. 8). 

The research objective (aim) is suggested to move to the introduction and add research questions. 

-Thank you for your valuable feedback. The chapter outlining the research aim has been removed, and the explanatory text has been repositioned to follow the literature review. 

Additionally, four hypotheses have been proposed, substantiated by the findings from the literature review as follows: 

H1: Emotional intelligence negatively affects burnout; 

H2: Emotional intelligence positively influences a sense of community and indirectly impacts trust between employees; 

H3: Sense of community and mutual trust between employees negatively affect burnout; 

H4: Workplace milieu resources mediate the relationship between emotional intelligence and burnout. 

Our theoretical model based on these assumptions is presented in Fig 1.

Research design is necessary for population, sampling technique, data collection, and data analysis (SPSS for descriptive analysis in frequency, percentage, and mean). 

-Thank you for the comment. Although the sampling procedure was already included in the original manuscript, it was incorrectly placed in the Participants section. We moved it to the Research Design and Sampling Procedures section and further supplemented it, as you can see at the beginning of the section (pp. 11-12).

Descriptive statistics have already been included in the Participant section, but we have supplemented this with additional informative indicators. The descriptive statistics of the relevant indicators are included at the beginning of the text of the Results section and in Table 1.

The inferential statistical analysis of PLS-SEM is necessary to explain.

-Thank you for the comment. The Data Analysis section was supplemented with a description of the serial mediation used in the study. We emphasized that we used a saturated model and stated that we used percentile bootstrap to estimate the standard errors and confidence intervals used for inferential statistics.

For data analysis in Table 2, please add *significant level at p<0.05, **significant level at p<0.01, and ***significant level at p<0.001.

-Thank you for the comment, we have added the asterisks and the notes to Table 2 and consequently to Table 3 and Table 4.

Discussions are needed to follow the research hypotheses

-Thank you for your insightful feedback regarding the discussion section. A concise summary of the results has been incorporated to align with the logical framework of the hypotheses.

More tables are required for assumptions.

-Thank you for the suggestion, the descriptive statistics used to check the assumptions have been tabulated and placed in the Supplementing information.

Avoid using we, but use the researchers instead. 

-Thank you for your feedback. The sentences that commenced with or included "we" have been rephrased for greater academic clarity.

When research hypotheses are unclear, discussions and conclusions are also not clear. The conclusion suggests how a strategic planner could improve the outcome based on predictors. 

-Research hypotheses have been incorporated, leading to a rephrasing of the discussion section to ensure alignment with these hypotheses. Additionally, future strategic policy implications and directions have been included to enhance the overall analysis.

Reviewer2

Reviewers' Comment Author(s) revision

When using Grammarly to check AI generated document similarity was found that 49% of this manuscript is similar to AI-generated documents. Thus, the reviewer suggest the authors revise the manuscript to be plagiarism and AI-similarity to be 15% or less. 

-We thank Reviewer 2 for his/her comment. We would like to inform Reviewer 2 that no original text was created using AI assistance. During the writing, sometimes we used DeepL and Google Translate for the translation (but always only short parts of about one sentence at a time were translated); we checked and corrected the grammar using Grammarly, and our article was proofread by the official English proofreading office of Semmelweis University (we can provide correspondence confirming this if required). However, we, the authors, are not native English speakers. Liang et al. (2023) found that AI detectors discriminate negatively against texts written by non-native English speakers (due to their more straightforward, predictable, rigid sentence structures). Therefore, while native English texts were judged relatively accurately by AI detectors, non-native texts had a high false positive rate. We think this may be the reason for the misunderstanding. 

Based on the comments of the other reviewers, the manuscript underwent a significant revision, during which we tried to rely even less on Google Translate and Grammarly, and we hope that the text will become acceptable. However, to further tweak the text to get a low hit value on the AI detector, we find this approach fundamentally dishonest. In addition, we added a paragraph at the end of the manuscript in which we stated the use of Google Translate, Grammarly, DeepL, and professional proofreading. We can only trust in the goodwill of Reviewer 2.

Liang et al (2023): https://arxiv.org/pdf/2304.02819

Reviewer3

Reviewers' Comment Authors revision

First suggestion, in the abstract there is no explanation of the methodology used, this is very important to be explained.

-Thank you for your statement we have extended and adjusted the abstract as required.

Second suggestion, to add an in-depth discussion of the problems and empirical gaps that occur so that this research has good useful value

-Thank you for your comment. We have added the following paragraph to the end of the introduction: 

"This study offers novel insights into how emotional intelligence influences burnout. It focuses on social care leaders in Hungary, a group vulnerable to burnout who also faces unique organizational and bureaucratic burdens in Eastern Europe. By highlighting workplace milieu resources as mediators, the research suggests that emotionally intelligent leaders are better equipped to use social resources at the workplace to mitigate burnout. Thus, the results of this study could further support tailored social policy interventions to improve social care leaders' mental health and working conditions."

The third suggestion, make hypothesis development on each influence relationship with relevant articles that support the research framework, because this research uses quantitative methods so it is mandatory to make hypotheses and will be answered in the discussion section. 

-Thank you for your valuable feedback. 

 Four hypotheses have been proposed, substantiated by the findings from the literature review as follows: 

H1: Emotional intelligence negatively affects burnout; 

H2: Emotional intelligence positively influences a sense of community and indirectly impacts trust between employees; 

H3: Sense of community and mutual trust between employees negatively affect burnout; 

H4: Workplace milieu resources mediate the relationship between emotional intelligence and burnout. 

Our theoretical model based on these assumptions is presented in Fig 1. The hypothesis is detailed and supported with literature in the new Hypothesis section (pages 9-10)

The fourth suggestion, in the methodology section, it is necessary to determine the sampling technique used, besides that to provide better value ... 

-Thank you for the comment. Although the sampling procedure was already included in the original manuscript, it was incorrectly placed in the Participants section - we moved it to the Research Design and Sampling Procedures section and further supplemented it, as you can see at the beginning of the section (p.11-12).

Researchers need to add questionnaire statements in tabular form so that they can be easily read. Thank you for your insight. In the appendix (supplementary documents), we have included a summary of all the survey questions analysed in a tabular format.

The fifth suggestion, in the research results section, there should be validity and reliability testing so that the data used can be relied on, at least the indicator part of each variable. -Thank you for the feedback, we would like to inform you that Cronbach's alphas, as a reliability indicator, were included in the Instruments section, after the presentation of all questionnaires. The reliability of the EI scales obtained as a result of PCA is also reported there. In order to avoid misunderstandings, we now emphasize in the text that the reliability values were calculated on the data of the present research. We also included the reliability of the EI scales in the table showing the results of the PCA in the Supporting Information. The purpose of this research is not to validate the AES questionnaire (it was validated in Hungarian language), we used PCA as a preliminary processing step, as a data reduction procedure, and therefore we only published its results in the Supporting Information, and thus the main text does not discuss the correlations of the EI scales with the other relevant variables as a validation process. However, it can be seen that the correlations of the scales were as expected.

The sixth suggestion, in the discussion section, it is highly recommended to discuss each influence relationship so that the results and discussion can be interpreted properly. -Thank you for your valuable insight. We have revised the discourse to reflect your feedback, ensuring all effects and logical connections emerged from our research are clearly identified, analysed and described 

The seventh suggestion, the conclusion should be first and then the implication, the conclusion is a summary of the discussion while the implication is the impact after this research is completed. 

-Thank you for your feedback. Based on your suggestion, we have adjusted the order of the sections in the text accordingly.

Reviewer4

Reviewers' Comment Authors revision

Limitations: The cross-sectional design limits the ability to infer causal relationships. This could be addressed in future research by incorporating longitudinal designs to observe changes over time. 

-Thank you for your insights. According to your suggestions, we have recreated and extended the limitations and future implications (in conclusion) section. (First limitation reflects the cross sectional design). In the future we are keen to make a longitudinal design as we are also interested in repeated measures SEM.

---

## [Decision Letter · Decision Letter 1]

7 Nov 2024

PONE-D-24-29991R1The mediating role of workplace milieu resources on the relationship between emotional intelligence and burnout among leaders in social carePLOS ONE

Dear Dr. Kozák,

Thank you for submitting your manuscript to PLOS ONE. After careful consideration, we feel that it has merit but does not fully meet PLOS ONE’s publication criteria as it currently stands. Therefore, we invite you to submit a revised version of the manuscript that addresses the points raised during the review process.

We look forward to receiving your revised manuscript.

Kind regards,

Supaprawat Siripipatthanakul, Ph.D.

Academic Editor

PLOS ONE

Journal Requirements:

Additional Editor Comments:

(1) The abstract is required for data collection (study setting of Hungary's population and sampling technique). Data analysis using statistical analysis for descriptive analysis (frequency, percentage, mean, and standard deviation) and hypothesis testing using inferential statistics are needed for justification.

(2) Please add all contents of the ethical statement in the methodology:

A comprehensive, multifaceted questionnaire was distributed to social care leaders at Semmelweis University in Budapest, Hungary, following ethical approval from the university's research review board (Ethical Approval of Research Semmelweis University Regional and Institutional Committee of Science and Research Ethics SE RKEB: 61/2019). Participants were asked to complete the questionnaire on paper at the conclusion of their training sessions. The questionnaire was in Hungarian, and participation was voluntary, with the option to withdraw at any time without consequences. Data was collected using anonymous pseudocodes. The recruitment period spanned from April 1, 2019, to November 30, 2019. During data collection, all adult participants enrolled in the social care management training at Semmelweis University were provided with a written consent form at the commencement of the questionnaire. Participation was voluntary and anonymous. They were informed that the processed results, containing anonymized individual data, would be published in scientific and educational publications.

(3) Kindly use the findings instead of our findings

(4) Kindly check Figure on page 49, b2 and d, if it needs to be replaced b2 to d and d to be b2.

Reviewers' comments:

Reviewer's Responses to Questions

**Comments to the Author**

1. If the authors have adequately addressed your comments raised in a previous round of review and you feel that this manuscript is now acceptable for publication, you may indicate that here to bypass the “Comments to the Author” section, enter your conflict of interest statement in the “Confidential to Editor” section, and submit your "Accept" recommendation.

Reviewer #1: All comments have been addressed

Reviewer #2: (No Response)

Reviewer #3: All comments have been addressed

Reviewer #4: All comments have been addressed

2. Is the manuscript technically sound, and do the data support the conclusions?

Reviewer #1: Yes

Reviewer #2: Yes

Reviewer #3: Yes

Reviewer #4: Yes

3. Has the statistical analysis been performed appropriately and rigorously? 

Reviewer #1: Yes

Reviewer #2: Yes

Reviewer #3: Yes

Reviewer #4: Yes

4. Have the authors made all data underlying the findings in their manuscript fully available?

Reviewer #1: Yes

Reviewer #2: Yes

Reviewer #3: Yes

Reviewer #4: Yes

5. Is the manuscript presented in an intelligible fashion and written in standard English?

Reviewer #1: Yes

Reviewer #2: Yes

Reviewer #3: (No Response)

Reviewer #4: Yes

6. Review Comments to the Author

Reviewer #1: Please check the abstract format of PLOS ONE and ensure it includes background, methods, results, and a conclusion. Methods in the abstract are also necessary for data collection and analysis.

Reviewer #2: Acceptable,

The revised version is acceptable. However, kindly check for plagiarism and AI Writing similarity to be less than 15%, if any, for typographical or grammatical errors.

Reviewer #3: (No Response)

Reviewer #4: I recommend increasing the introduction and literature review to include fully defined variables and more detailed hypothesis development based on relevant research to strengthen the study's foundation. The introduction should state the research purpose and provide specific research questions to help readers comprehend the study's goals. Clarifying sampling, statistical analysis, and inferential approaches (e.g., PLS-SEM and JASP for serial mediation analysis) in the methods section would improve transparency. Adding a methodologies and analytic approaches overview to the abstract will help set reader expectations. Validity and reliability testing for measurement tools and better significance indications in tables would strengthen results. The explanation should structure the analysis around the hypotheses, detail each relationship, and address strategic planner implications to improve interpretability and application. Though Reviewer 2 appreciates the clarity on translation and proofreading processes, minimizing automated tool use may minimize AI-similarity concerns. Reviewer 4's recommendation to acknowledge cross-sectional design limitations and propose longitudinal research could improve the study's causal inferences. These changes would improve the manuscript's fit with journal standards and add to the knowledge on emotional intelligence, burnout, and social care workplace resources.

7. PLOS authors have the option to publish the peer review history of their article (what does this mean?). If published, this will include your full peer review and any attached files.

Reviewer #1: No

Reviewer #2: No

Reviewer #3: No

Reviewer #4: No

---

## [Author Response · Author response to Decision Letter 1]

6 Dec 2024

Dear Editor and Reviewres,

Please see the comments aligned with the Authors revisions.

(1) The abstract is required for data collection (study setting of Hungary's population and sampling technique). Data analysis using statistical analysis for descriptive analysis (frequency, percentage, mean, and standard deviation) and hypothesis testing using inferential statistics are needed for justification.

-Thank you very much for your comment. We have revised and rewritten the abstract according to the required format, added information and suggested subsections (also aligned with the comments of Reviewers 1 & 4)

(2) Please add all contents of the ethical statement in the methodology:

A comprehensive, multifaceted questionnaire was distributed to social care leaders at Semmelweis University in Budapest, Hungary, following ethical approval from the university's research review board (Ethical Approval of Research Semmelweis University Regional and Institutional Committee of Science and Research Ethics SE RKEB: 61/2019). Participants were asked to complete the questionnaire on paper at the conclusion of their training sessions. The questionnaire was in Hungarian, and participation was voluntary, with the option to withdraw at any time without consequences. Data was collected using anonymous pseudocodes. The recruitment period spanned from April 1, 2019, to November 30, 2019. During data collection, all adult participants enrolled in the social care management training at Semmelweis University were provided with a written consent form at the commencement of the questionnaire. Participation was voluntary and anonymous. They were informed that the processed results, containing anonymized individual data, would be published in scientific and educational publications.

-Thank you for your comment we have restructured the "Research design and sampling procedures" & "Ethics Statement" sections and added all the required information.

(3) Kindly use the findings instead of our findings

-Thank you for your comment. We have replaced “our “findings with “the” findings.

(4) Kindly check Figure on page 49, b2 and d, if it needs to be replaced b2 to d and d to be b2.

-Thank you for the suggestion. We checked the labelling, and it was as we intended. Our reasoning is: we wanted to keep the labelling design commonly used in mediation analyses, i.e. regression lines originating from the predictor and leading to the mediator(s) are commonly labelled with “a”, and regression lines originating from the mediators and leading to the outcome variable are commonly labelled as “b”. Finally, the regression line between the two mediators is labelled as “d”. This way, a1 and b1 are the lines leading to and originating from the first mediator, and a2 and b2 are connected to the second mediator.

Authors Comments to the Reviewers

Reviewer1

Reviewers' Comment Authors revision

Please check the abstract format of PLOS ONE and ensure it includes background, methods, results, and a conclusion. Methods in the abstract are also necessary for data collection and analysis.

-Thank you very much for your comment. We have revised and rewritten the abstract according to the required format and suggested subsections (also aligned with the comments of the Editor & Reviewer 4)

Reviewer2

Reviewers' Comment Author(s) revision

The revised version is acceptable. However, kindly check for plagiarism and AI Writing similarity to be less than 15%, if any, for typographical or grammatical errors. 

-Thank you for your comment. We have re-verified the information with https://detecting-ai.com/, which indicated a rate of 6.8%, Grammarly 7% while justdone.ai suggested 74%. We would like to emphasize that we are not native speakers and therefore employed Grammarly, DeepL, and support from the university's lecture office for grammar and semantic checks. However, we wish to clarify that no AI was used to generate ideas or compose the text. 

Reviewer4

Reviewers' Comment Authors revision

recommend increasing the introduction and literature review to include fully defined variables and more detailed hypothesis development based on relevant research to strengthen the study's foundation. The introduction should state the research purpose and provide specific research questions to help readers comprehend the study's goals. 

-Thank you for your valuable feedback. We have revised, rewrote and reorganized the Introduction and Literature Review sections to enhance clarity, flow, and detail, ensuring that all variables are fully defined. This provides a solid foundation for the Research Objectives and Hypothesis Development section. In the introduction, we emphasized the research purpose and included questions to guide the reader more effectively.

Clarifying sampling, statistical analysis, and inferential approaches (e.g., PLS-SEM and JASP for serial mediation analysis) in the methods section would improve transparency 

-Thank you for the suggestions, we expanded the Methods section with details of sampling and descriptive statistics. The final version of the description of the mediations are as follows:

„…the serial mediation models included only manifest variables calculated from the questionnaire responses (commonly called Path Analysis or the structural part of CB-SEM) and they were examined in a saturated model, which is one of the accepted approaches in the field of the current research []. In the saturated model, the point estimates of the regression coefficients and the indirect effects are the same, regardless of which regression calculation method or structural equation modelling is used (in this study, for reasons of convenience, Maximum Likelihood modelling was used). Standard Errors and Confidence intervals were obtained by percentile bootstrap for regression coefficients and indirect effects as well. Lavaan syntax of the saturated mediation models can be found in S3.”

We hope that this description adequately expresses that we did not use PLS-SEM, but a CB-SEM, in which we tested a saturated model and used the variables in manifest form, with their scale scores calculated in advance (i.e. we used a structural model). The reliability of the questionnaires was described during the presentation of the questionnaires but we also added the Cronbach’s alpha values to Table 1.

Adding a methodologies and analytic approaches overview to the abstract will help set reader expectations. 

-Thank you very much for your comment. We have revised and rewritten the abstract according to the required format and suggested subsections (also aligned with the comments of the Editor & Reviewer 1)

Validity and reliability testing for measurement tools and better significance indications in tables would strengthen results. 

-Thank you for the suggestion. Reliability measures (Cronbach’s alphas) are reported in the Methods section after the description of each questionnaire. However, we also included these values to the correlation table. We hope that this solution will meet the desired reporting standards.

Significance is indicated with p-values, with asterisks, bolding, and confidence intervals. Now we also mentioned in the notes the meaning of bold. We struggle to find any better significance indications. 

The explanation should structure the analysis around the hypotheses, detail each relationship, and address strategic planner implications to improve interpretability and application. 

-Thank you for your insightful comment. We have provided specific references to the relevant lines for each hypothesis in the hypotheses section. We have mirrored this approach in the discussion section, ensuring that all relationships are thoroughly articulated. 

-Thank you very much for your comment. According to the comments, we have rewritten, restructured, and extended the section, which is newly named “Theoretical Implications and Limitations.”

Though Reviewer 2 appreciates the clarity on translation and proofreading processes, minimizing automated tool use may minimize AI-similarity concerns. 

-Thank you for your comment. We have re-verified the information with https://detecting-ai.com/, which indicated a rate of 6.8%, Grammarly suggested 7% while justdone.ai suggested 74%. We would like to emphasize that we are not native speakers and therefore employed Grammarly, DeepL, and support from the university's lecture office for grammar and semantic checks. However, we wish to clarify that no AI was used to generate ideas or compose the text. 

Reviewer 4's recommendation to acknowledge cross-sectional design limitations and propose longitudinal research could improve the study's causal inferences. 

-Thank you very much for your comment we extended the Theoretical implications and Limitations as follows: “Firstly, this study was based on a cross-sectional design. As a result, it cannot reflect the evolving dynamics and interactions between the studied phenomena, and reverse or reciprocal causal relationships cannot be analysed. To address these limitations, future research could employ a longitudinal design to examine changes and interactions among emotional intelligence, workplace resources, and burnout over time.”

---

## [Decision Letter · Decision Letter 2]

18 Dec 2024

PONE-D-24-29991R2The mediating role of workplace milieu resources on the relationship between emotional intelligence and burnout among leaders in social carePLOS ONE

Dear Dr. Kozák,

Thank you for submitting your manuscript to PLOS ONE. After careful consideration, we feel that it has merit but does not fully meet PLOS ONE’s publication criteria as it currently stands. Therefore, we invite you to submit a revised version of the manuscript that addresses the points raised during the review process.

We look forward to receiving your revised manuscript.

Kind regards,

Supaprawat Siripipatthanakul, Ph.D.

Academic Editor

PLOS ONE

Journal Requirements:

Additional Editor Comments :

Acceptable; however, the recruitment period spanned from April 11th to November 30th, 2019. The results are reflected in 2019 (not 2024 as of the current year). If COVID-19 may affect the results of the study "The Mediating Role of Workplace Milieu Resources on the Relationship between Emotional Intelligence and Burnout among Leaders in Social Care," Please discuss whether further studies are recommended in this inquiry

Reviewers' comments:

Reviewer's Responses to Questions

**Comments to the Author**

1. If the authors have adequately addressed your comments raised in a previous round of review and you feel that this manuscript is now acceptable for publication, you may indicate that here to bypass the “Comments to the Author” section, enter your conflict of interest statement in the “Confidential to Editor” section, and submit your "Accept" recommendation.

Reviewer #1: All comments have been addressed

2. Is the manuscript technically sound, and do the data support the conclusions?

Reviewer #1: Yes

3. Has the statistical analysis been performed appropriately and rigorously? 

Reviewer #1: Yes

4. Have the authors made all data underlying the findings in their manuscript fully available?

Reviewer #1: Yes

5. Is the manuscript presented in an intelligible fashion and written in standard English?

Reviewer #1: Yes

6. Review Comments to the Author

Reviewer #1: Acceptable; however, the recruitment period spanned from April 11th to November 30th, 2019. The results are reflected in 2019 (not 2024 as of the current year). If COVID-19 may affect the results of the study "The Mediating Role of Workplace Milieu Resources on the Relationship between Emotional Intelligence and Burnout among Leaders in Social Care," Please discuss whether further studies are recommended in this inquiry.

7. PLOS authors have the option to publish the peer review history of their article (what does this mean?). If published, this will include your full peer review and any attached files.

Reviewer #1: No

---

## [Author Response · Author response to Decision Letter 2]

23 Dec 2024

Editor & Reviewer Comment:Acceptable; however, the recruitment period spanned from April 11th to November 30th, 2019. The results are reflected in 2019 (not 2024 as of the current year). If COVID-19 may affect the results of the study "The Mediating Role of Workplace Milieu Resources on the Relationship between Emotional Intelligence and Burnout among Leaders in Social Care," Please discuss whether further studies are recommended in this inquiry

Author response:Thank you for this valuable observation. Indeed, the interventions to tackle the COVID-19 pandemic, may have exacerbated burnout and impacted emotional intelligence among social care leaders. However, research on this subject is sparse in the local setting. The authors decided to reflect on this observation by revising the "Theoretical Implications and Limitations section" to add a final limitation paragraph of the manuscript. We hope that the added text meets the Editor's and the Reviewer's expectations.

The added limitation paragraph is as follows: 

" The recruitment period for this study (April 11th to November 30th, 2019) predates the global COVID-19 pandemic, which has drastically reshaped workplace dynamics, particularly in the social and healthcare sectors. Although Hungary largely resumed in-person work—aligned with the personal nature of social work—the pandemic introduced profound challenges that could exacerbate burnout and workplace demands. For example, Dima, Schmitz, and Simon's (2021) research found elevated stress levels among Romanian social workers during the pandemic associated with increased burnout (72). Whereas McCoyd and Colleagues suggested that concerning the COVID-19 pandemic, social workers demonstrated resilience, adapting to increased demands, integrating new technologies, and addressing service gaps despite experiencing symptoms of burnout (73). These findings suggest that the structural and psychological impacts of the pandemic may have influenced the interplay between emotional intelligence, workplace resources, and burnout prevention among leaders in social care. A follow-up study incorporating post-pandemic data will address these limitations, focusing on workplace demands, structural changes, and the evolving roles of emotional intelligence and resources in mitigating burnout. This future research aims to understand better the social care sector's changing dynamics and practical strategies for leader support in a post-pandemic context.”

---

## [Editor Report · Decision Letter 3]

26 Dec 2024

The mediating role of workplace milieu resources on the relationship between emotional intelligence and burnout among leaders in social care

PONE-D-24-29991R3

Dear Dr. Kozák,

We’re pleased to inform you that your manuscript has been judged scientifically suitable for publication and will be formally accepted for publication once it meets all outstanding technical requirements.

Kind regards,

Supaprawat Siripipatthanakul, Ph.D.

Academic Editor

PLOS ONE

Additional Editor Comments (optional):

Grammar is OK, plagiarism is 1%, and AI writing similarity of 13% is fine. The revised version is overall acceptable regarding its content and analysis.
---

## [Editor Report · Acceptance letter]

7 Jan 2025

PONE-D-24-29991R3 

PLOS ONE

Dear Dr. Kozák, 

I'm pleased to inform you that your manuscript has been deemed suitable for publication in PLOS ONE. Congratulations! Your manuscript is now being handed over to our production team.

Kind regards, 

on behalf of

Professor Supaprawat Siripipatthanakul 

Academic Editor

PLOS ONE